# Molecular motor-driven reversible liquid-liquid phase separation of supramolecular assemblies

Fan Xu [1,3], Marco Ovalle[1,4], Youxin Fu[1,5], Marc A. C. Stuart [2] & Ben L. Feringa [1] ✉

Liquid-liquid phase separation (LLPS) is a crucial process in natural and artificial systems, capable of maintaining cellular behavior and realizing material functions. While supramolecular assemblies provide a versatile platform for understanding natural phenomena and developing adaptive materials, their LLPS remains largely unexplored, particularly with respect to reversible control. Here, we report a molecular motor-driven LLPS system, where nanoscale rotary motion modulates LLPS of supramolecular assemblies. Systematic molecular modification and photothermal isomerization studies comprehensively reveal that subtle changes in molecular structure affect the hydrophobicity of molecules, which in turn decrease the critical phase separation temperature and promotes the phase separation. During the rotary of molecular motor, these assemblies undergo in situ formation and dissolution of droplets across multiple non-equilibrium states. Our findings establish an orthogonal strategy to tune phase separation by light and temperature, providing an avenue for designing out-of-equilibrium biomedical materials and adaptive soft matter systems.

Liquid-liquid phase separation (LLPS) is a prevalent phenomenon observed in natural systems, such as forming membraneless organelles, playing a critical role in the functioning and behavior of cells[1-10]. Biomolecular condensates formed through LLPS are associated with neurodegenerative diseases[11-13], regulation of viral assembly[14], and great therapeutic potential[15,16]. In the realm of artificial materials, synthetic polymers were studied to understand the mechanism and to achieve functionality through LLPS[17-26]. For instance, polymer coacervates are regarded as the models of membranelles organelles[27-30] and protocells[31-33], enabling biomimetic functions[31], such as predation[32] and amplification of DNA[33].

Compared to biomacromolecules and synthetic polymers, supramolecular assemblies of small molecules (SASMs) leverage non-covalent interactions (e.g., hydrogen bonding and π-π stacking), offering enhanced dynamics and reversibility[34-38]. When the assemblies are one-dimensional ordered structures, they share many characteristics with polymers and become supramolecular polymers[39-41]. Supramolecular assembly of synthetic small molecules represents an innovative approach that not only provides valuable insights into intricate biological processes such as molecular recognition, but also results in adaptive systems with easily adjustable functionality[34-36,42,43]. However, compared to biomacromolecule condensates and polymer coacervates[44-51], only a few explorations have been undertaken into LLPS of supramolecular polymers and assemblies of small molecules[52-54].

A crucial challenge in LLPS research is achieving precise and reversible control over phase separation for adaptive functionality[55,56]. LLPS has been shown to achieve responsiveness with different stimuli, such as thermal[57,58] and charge intensity[47]. For instance, RNAs and

[1]Synthetic Organic Chemistry, Stratingh Institute for Chemistry, University of Groningen, Groningen, AG, The Netherlands. [2]Groningen Biomolecular Sciences and Biotechnology Institute, University of Groningen, Groningen, AG, The Netherlands. [3]Present address: Institute for Complex Molecular Systems and Laboratory of Macromolecular and Organic Chemistry, Eindhoven University of Technology, Eindhoven, The Netherlands. [4]Present address: IMDEA Nanociencia, Madrid, Spain. [5]Present address: College of Science, Nanjing Forestry University, Nanjing, China. ✉e-mail: b.l.feringa@rug.nl

proteins undergo phase transitions when increasing the temperature above their lower critical solution temperatures (LCST)[57,58]. Among various stimuli, light is privileged due to its remote action with high spatial-temporal precision[59,60]. There is growing interest in achieving reversible control of LLPS using light[19,31,61]. The molecular switch azobenzene has been utilized as a photoresponsive amphiphile to modulate the formation of polymer coacervates[19,31], and has also been covalently conjugated to DNA to drive life-like motion in water/oil droplet systems[61].

In contrast to two-state molecular switches, light-driven molecular motors based on overcrowded alkenes are artificial molecular machines enabling unidirectional rotation[62], and have been introduced to control dynamic functions[63–65], such as actuation[66–69] and cell membrane disruption[70]. We hypothesize that the directional rotation of molecular motor induces sequential structural changes among its four isomeric states, thereby enabling precise, multistate control over LLPS. Such sequential modulation of phase behavior may facilitate the execution of complex tasks in biomedical applications.

In the present study, we develop a molecular motor-driven LLPS of supramolecular assemblies, orthogonally coupled with temperature control (Fig. 1). A key feature of our approach is the use of a single molecular component to achieve multistate control over LLPS, leveraging directional and sequential control capabilities that are not accessible with traditional molecular switches. Continuous photoisomerization and thermal helix inversion (THI) of molecular motors enable sequential and reversible variations in dipole moment and hydrogen bonding, which further contribute to different critical phase separation temperatures ($T_c$) and thus enable the in situ phase separation. $T_c$ can be precisely manipulated by adjusting the length of hydrophilic units, covering a temperature range from 18 to 52 °C. Our findings establish a molecular design principle for modulation of LLPS, highlighting sequential multistate phase separation control via molecular motor rotation and providing a orthogonal control strategy for future biomedical applications involving out-of-equilibrium capture and release functions.

## Results

### Molecular design and rotary motion of molecular motors

Molecular motors perform unidirectional 360° rotation, comprising two photoisomerization steps and two THI steps. The mechanism of photoisomerization involves light-induced excitation from the ground state to an excited state, triggering rotation around the central C=C double bond and resulting in $E/Z$ isomerization. In the subsequent THI step, the molecule undergoes a thermally activated helical inversion, driven by steric strain and thermal energy. During the rotation, four states can be observed: two stable states and two metastable states (Fig. 1a). The rotational speed of molecular motors can be tuned by molecular design[64]. To achieve rapid responsiveness in LLPS regulation, amphiphiles were designed based on a second-generation molecular motor, exhibiting a half-life of seconds as monomers and extending to minutes within aqueous assemblies. Bis-urea groups relate to the molecule motors through C3 alkyl chains and are linked with hydrophilic oligoethylene glycol moieties through C6 alkyl chains, thus enabling urea groups in the hydrophobic pockets, which should offer hydrogen bonds for supramolecule assemblies. The oligoethylene glycol chains, comprising triethylene glycol (OEG3), tetraethylene glycol (OEG4), and hexaethylene glycol (OEG6), were incorporated to modulate the $T_c$. The synthesis of second-generation molecular motor amphiphiles **2MOEG6, 2MOEG4**, and **2MOEG3** was described in the Supplementary Information, and all the molecules were characterized by $^1$H, $^{13}$C NMR, and high-resolution MS (Supplementary Figs. 1, 28–37 and 39–41).

The rotary process of molecular motors was investigated through the utilization of $^1$H NMR spectroscopy. Upon irradiation with 365 nm light for 70 min at −20 °C, proton signals of $H_a$ ($\delta$ = 6.89 ppm) and $H_b$ ($\delta$ = 2.62 ppm) shift downfield to 7.30 and 2.84 ppm, respectively (Fig. 2a), indicative of a conversion from the stable $Z$ isomer ($Z_S$-**2MOEG4**) to the metastable $E$ isomer ($E_M$-**2MOEG4**). The ratio of $E_M$-**2MOEG4** to $Z_S$-**2MOEG4** is 73:27 by integrating the $^1$H NMR signals at the photostationary state (PSS). Subsequently, upon warming the sample at 20 °C in the dark for 7 min, the proton signal of $H_a$ exhibits a downfield shift, while the proton signal of $H_b$ shifts upfield, which is in accordance with the transition of $E_M$-**2MOEG4** to stable $E$ isomer ($E_S$-**2MOEG4**) with full conversion. Irradiating the mixture with 365 nm light for 90 min at −20 °C results in an upfield shift of $H_a$ and downfield shift of $H_b$, indicating the photoisomerization of $E_S$-**2MOEG4** to metastable $Z$ isomer ($Z_M$-**2MOEG4**) with a ratio of 76:24 at the PSS. After subsequent warming at 20 °C for 7 min, the proton signal of $H_a$ shifts downfield while $H_b$ shifts highfield, indicating the THI of $Z_M$-**2MOEG4** to $Z_S$-**2MOEG4**. The final mixture after a rotation cycle contains $Z_S$-**2MOEG4** and $E_S$-**2MOEG4** with a ratio of 60:40. The rotation behavior was further monitored by time-dependent UV-Vis absorption spectroscopy at −3 °C. Upon irradiation, the absorption band at 300–385 nm of $Z_S$-**2MOEG4** exhibits a decrease, accompanied by the emergence of a new band at 386–485 nm, indicating

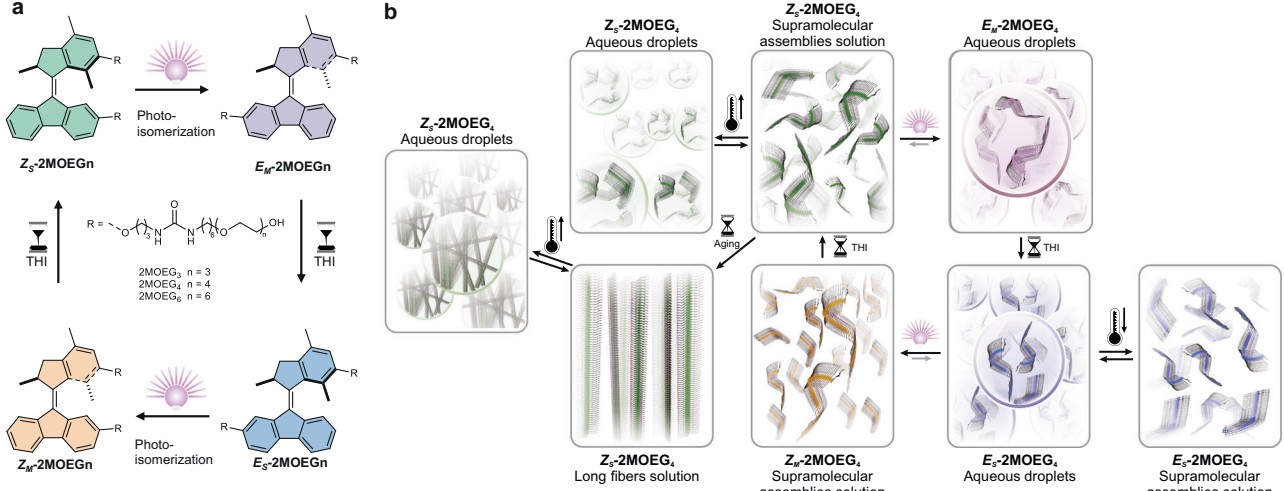

**Fig. 1 | Molecular motor-driven multi-state liquid-liquid phase separation of supramolecular assemblies. a** Four-state rotary of second-generation molecular motor amphiphiles. **b** Orthogonal control of liquid–liquid phase separation by light and temperature in aqueous solutions of molecular motor assemblies. THI: thermal helix inversion, bulb: the irradiation of light, hourglass: the passage of time, thermometer: the change in temperature.

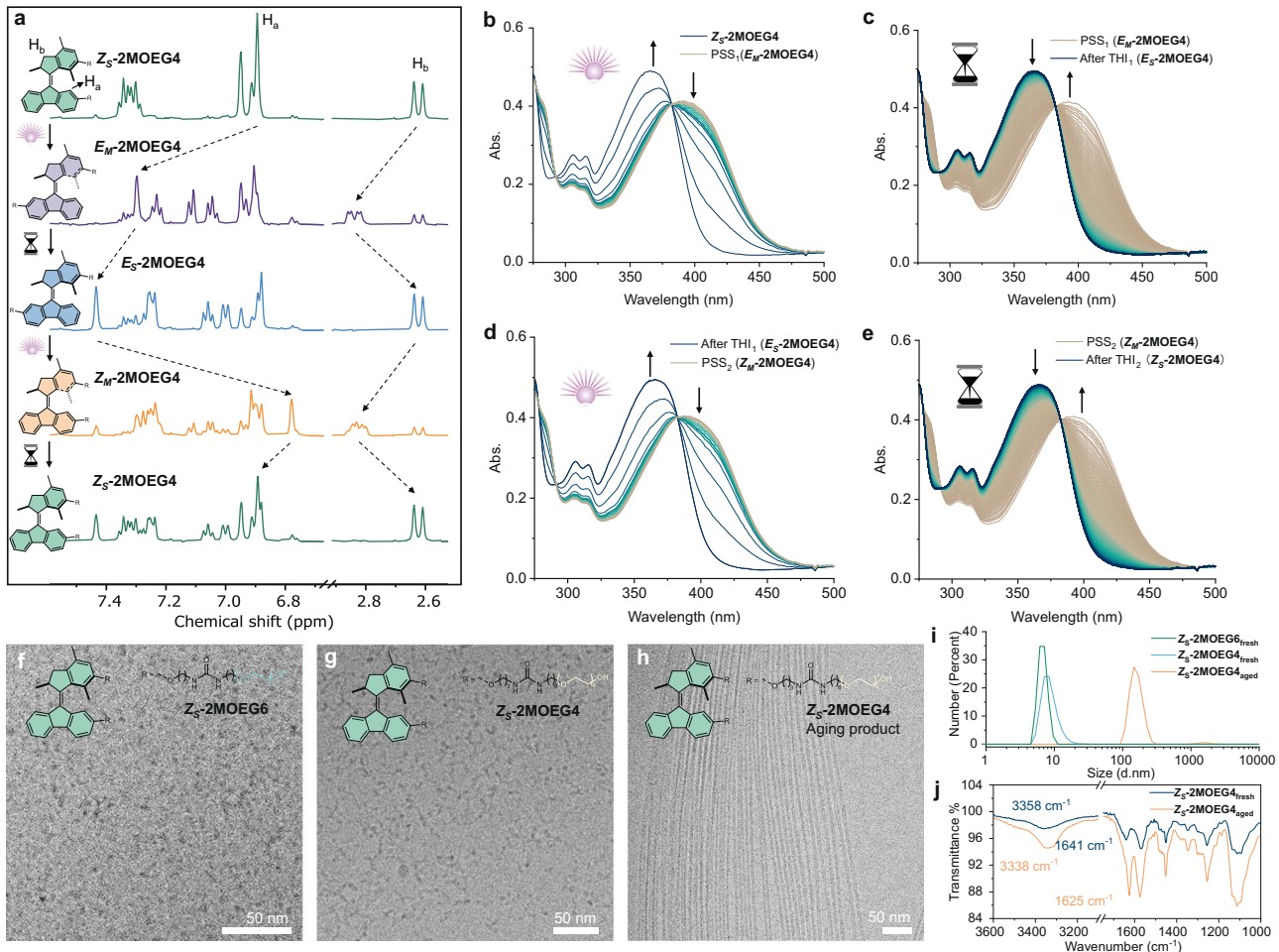

**Fig. 2 | Rotation and assembly of molecular motors. a** $^1$H NMR (500 MHz, CD$_3$OD) stacked spectra (from top to bottom) of pristine $Z_S$-**2MOEG4** after 365 nm light irradiation for 70 min to get $E_M$-**2MOEG4** at −20 °C. Then the sample was maintained in the dark at 20 °C for 7 min to reach $E_S$-**2MOEG4**. Subsequent irradiation with 365 nm light at −20 °C for 90 min yields $Z_M$-**2MOEG4**. Finally, the sample was kept in the dark at 20 °C for 7 min to recover $Z_S$-**2MOEG4**. **b**–**e** UV-Vis absorption spectra of **b** $Z_S$-**2MOEG4** (50 μM, MeOH, −3 °C) upon 365 nm light irradiation for 10 min to PSS$_1$ getting $E_M$-**2MOEG4** followed by **c** keeping in the dark for 60 min to process THI$_1$ and reach $E_S$-**2MOEG4**; **d** subsequent irradiating with 365 nm light for 20 min to PSS$_2$ getting $Z_M$-**2MOEG4** followed by **e** maintaining in the dark for 60 min to process THI$_2$ and recover $Z_S$-**2MOEG4**. Cryo-TEM images of assembled morphologies of **f** $Z_S$-**2MOEG6**, **g** $Z_S$-**2MOEG4**, and **h** aged $Z_S$-**2MOEG4** in water. **i** Dynamic light scattering data of $Z_S$-**2MOEG6**, $Z_S$-**2MOEG4**, and aged $Z_S$-**2MOEG4** assemblies in water. **j** FTIR spectra of the $Z_S$-**2MOEG4** assembly and the same sample after aging in water for one week. PSS: photostationary state, THI: thermal helix inversion, bulb: the irradiation of light, hourglass: the passage of time.

the formation of $E_M$-**2MOEG4** (Fig. 2b). After light irradiation of 10 min, the system reaches the first photostationary state (PSS$_1$). Subsequently keeping the sample in the dark for 60 min resulted in the THI of $E_M$-**2MOEG4** to $E_S$-**2MOEG4** (THI$_1$) (Fig. 2c). A subsequent irradiation of $E_S$-**2MOEG4** resulted in an increase in the absorption band at 386–485 nm until the second photostationary state (PSS$_2$), indicating the formation of $Z_M$-**2MOEG4** (Fig. 2d). Finally, the band at 386–485 nm disappeared with an increase of absorption band at 300-385 nm when the sample was kept in the dark, indicating the THI of $Z_M$-**2MOEG4** to $Z_S$-**2MOEG4** (THI$_2$) (Fig. 2f). Isosbestic points at 386 nm were observed in all the processes, which indicates that the sole process occurred in each step. The results are consistent with those obtained from NMR analysis. **2MOEG6** showed the same rotary behavior as **2MOEG4** (Supplementary Fig. 2). In summary, $^1$H NMR and UV-Vis studies reveal that motors undergo unidirectional rotation, exhibiting two metastable and two stable states.

## Supramolecular assemblies of molecular motors
Cryogenic transmission electron microscopy (cryo-TEM) study revealed that molecular motor $Z_S$-**2MOEG6** formed micelles with a diameter of 5 nm in water (Fig. 2f), while $Z_S$-**2MOEG4** formed worm-like micelles with a length of approximately 20-50 nm and a

comparable diameter to $Z_S$-**2MOEG6** (Fig. 2g). $Z_S$-**2MOEG3** was too hydrophobic to dissolve in water at room temperature (RT). The assembly morphology was found to be consistent with the packing parameters theory ($P = V/a_0 l_c$), where the smaller hydrophilic head groups ($a_0$) resulted in larger packing parameters $P$[71], and the increasing parameters resulted in the morphology transforming from micelles to worm-like micelles. Therefore, $Z_S$-**2MOEG6** showed micelles ($P < 1/3$) while $Z_S$-**2MOEG4** showed worm-like micelles ($1/3 < P < 1/2$) due to the shorter chain length of **OEG** that has a smaller area of hydrophilic head groups. Interestingly, $Z_S$-**2MOEG4** were found to form fibers with a uniform diameter of 5 nm and lengths of several micrometers after one week of aging (Fig. 2h). Dynamic light scattering (DLS) measurements revealed that the micelles of $Z_S$-**2MOEG6** showed a hydrodynamic diameter of around 6 nm. $Z_S$-**2MOEG4** showed a hydrodynamic diameter of around 7 nm, whereas the aged sample of $Z_S$-**2MOEG4** showed a hydrodynamic diameter of 150 nm. FTIR revealed a N–H stretching band at 3338 cm$^{-1}$ and a C=O stretching band at 1625 cm$^{-1}$ of aged $Z_S$-**2MOEG4** (straight fibers), corresponding to approximately 20 cm$^{-1}$ shifts compared to the fresh assembly (Fig. 2j). These shifts suggest enhanced urea hydrogen bonding upon aging, supporting the transformation from worm-like micelles to the

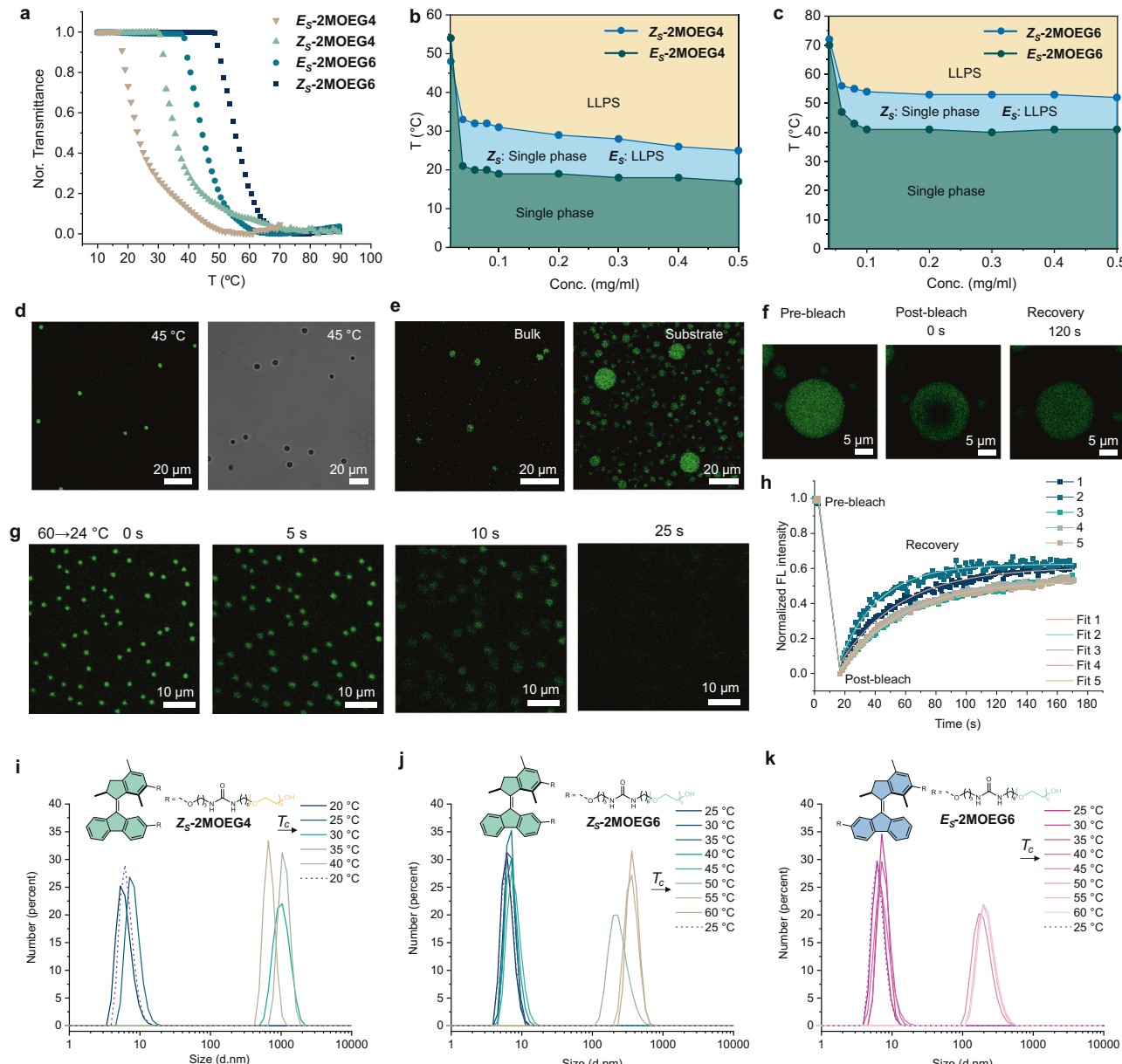

**Fig. 3 | Thermal-responsive phase separation of aqueous solutions of molecular motor assemblies. a** Transmittance of aqueous solutions of $E_S$-2MOEG4, $Z_S$-2MOEG4, $E_S$-2MOEG6, and $Z_S$-2MOEG6 upon heating. **b** Phase diagram of $E_S$-2MOEG4 and $Z_S$-2MOEG4 (0.02–0.5 mg/ml) in water. The $T_c$ was plotted as a function of concentrations. Green area: single-phase; yellow area: liquid–liquid phase separation (LLPS); blue area: $Z$ isomers form a single phase, whereas $E$ isomers undergo LLPS. **c** Phase diagram of $E_S$-2MOEG6 and $Z_S$-2MOEG6 (0.04–0.5 mg/ml) in water. **d** CLSM (left) and bright-field microscopic (right) images of $Z_S$-2MOEG4

droplets at 45 °C. **e** $E_S$-2MOEG4 droplets in the bulk solution and the glass substrate. **f** Time-dependent CLSM images of $E_S$-2MOEG4 droplets in FRAP measurements. **h** Corresponding kinetics of fluorescence recovery of five $E_S$-2MOEG4 droplets. **g** Time-lapse CLSM images of $Z_S$-2MOEG4 droplets after cooling from 60 (above $T_c$) to 24 °C (below $T_c$). $T_c$: critical phase separation temperature. Temperature-dependent DLS data of **i** $Z_S$-2MOEG4, **j** $Z_S$-2MOEG6 and **k** $E_S$-2MOEG6 in aqueous solutions.

more ordered micrometer-scale straight fibers. Time-dependent DLS measurements revealed that $Z_S$-2MOEG4 transformed into the fibers in one day (Supplementary Fig. 7), while micelles of $Z_S$-2MOEG6 remained the same after one week (Supplementary Fig. 8). $Z_S$-2MOEG6 is more hydrophilic than $Z_S$-2MOEG4 due to its longer hydrophilic **OEG** chains, which may reduce the hydrophobic interactions necessary for forming more ordered assemblies upon aging.

**Thermal-responsive liquid-liquid phase separation**

The critical phase separation temperature ($T_c$) of stable isomers of **2MOEG4** and **2MOEG6** was investigated by temperature-dependent transmittance measurements (Supplementary Information for details,

Supplementary Fig. 5). The aqueous solutions of $E_S$-2MOEG4, $Z_S$-2MOEG4, $E_S$-2MOEG6, and $Z_S$-2MOEG6 showed increased $T_c$ (Fig. 3a). The higher $T_c$ of **2MOEG6** compared to **2MOEG4** is due to the longer **OEG** chains with improved hydrophilicity of the molecules[72]. $E$ isomers of both molecular motors showed lower $T_c$ than their $Z$ isomers, possibly due to the greater hydrophobicity of the $E$ isomer compared to the $Z$ isomer[73]. We further plotted the phase diagram of these two molecular motors with their isomers. The $T_c$ of $Z_S$-2MOEG4 tended to decrease up to 0.1 mg/mL and then stabilized at around 29 °C as the concentration increased, whereas $T_c$ of $E_S$-2MOEG4 stabilized at 18 °C (Fig. 3b). $T_c$ of $Z_S$-2MOEG6 tended to decay until 0.1 mg/mL and then stabilized at 52 °C as concentration increased, whereas $T_c$ of $E_S$-

**2MOEG6** stabilized at 41 °C (Fig. 3c). FTIR analysis showed N–H stretching at 3358 cm$^{-1}$ for both $Z_S$ and $E_S$ assemblies, indicating hydrogen-bonded urea moieties (Supplementary Fig. 11). A minor redshift in the C=O stretching band from 1641 cm$^{-1}$ ($Z_S$) to 1637 cm$^{-1}$ ($E_S$) suggests slightly stronger hydrogen bonding in $E_S$-**2MOEG4**. Nile red fluorescence assay was used to probe the internal hydrophobicity of assemblies. The critical aggregation concentration (CAC) of $Z_S$-**2MOEG4** is 2.0 μM, higher than the 0.5 μM of $E_S$-**2MOEG4**, suggesting greater hydrophobicity of $E$ isomer (Supplementary Fig. 12). This result is consistent with dipole moment calculations performed using density functional theory (DFT), which show that the $Z_S$-**2MOEG4** exhibits a dipole moment of 9.65 D, while the $E_S$-**2MOEG4** has a lower value of 6.49 D, indicating a lower polarity for the $E$ isomer (Supplementary Information for details, Supplementary Fig. 26 and Supplementary Data 1). Notably, the $T_c$ of the aged $Z_S$-**2MOEG4** assembly solution decreased by 4 °C at the same concentration (Supplementary Fig. 6). The decrease of $T_c$ may be attributed to the increased length of supramolecular polymers upon aging[74].

Confocal laser scanning microscopy (CLSM) was employed to investigate LLPS. The aqueous solution $Z_S$-**2MOEG4** displayed homogeneity at 24 °C (Supplementary Fig. 13). At 45 °C (above $T_c$), the aqueous solution exhibited the formation of drops after phase separation (Fig. 3d). Frames from time-lapse movies of CLSM (Supplementary Movie 1) revealed that the droplets rapidly dissolved when the temperature decreased below the $T_c$ (Fig. 3g). Upon reaching a temperature exceeding the $T_c$, the generation of droplets was observed again (Supplementary Fig. 14, Supplementary Movie 2). The aqueous solution of $E_S$-**2MOEG4** has been found to perform LLPS at RT (Fig. 3b). Droplets of $E_S$-**2MOEG4** were observed in the bulk solution and glass substrates (Fig. 3e). The droplets in the bulk solution exhibited rapid motion (Supplementary Movie 3). To elucidate the internal dynamics of droplets, we performed fluorescence recovery after photobleaching (FRAP) measurements of droplets on the substrate. After photobleaching, $E_S$-**2MOEG4** droplets exhibited a rapid recovery of fluorescence signals, with a half-life time ($t_{1/2\ FRAP}$) of 27.0 ± 5.3 s (Fig. 3f, h), indicating the liquid-like feature of droplets. DLS studies showed that the hydrodynamic diameter of the droplets of $Z_S$-**2MOEG4** was around 1 μm after phase separation (Fig. 3i). The hydrodynamic diameter of the droplets formed by aged $Z_S$-**2MOEG4** assemblies is around 400 nm upon phase separation (Supplementary Fig. 10). The hydrodynamic diameter of the $Z_S$-**2MOEG6** droplets was approximately 360 nm (Fig. 3j), while the droplets of $E_S$-**2MOEG6** exhibited a diameter of about 200 nm after phase separation (Fig. 3k). Notably, when the temperature decreases below $T_c$, all types of assemblies returned to their original sizes, confirming the reversibility of the LLPS process (Fig. 3i-k and Supplementary Fig. 10). DLS data revealed the size of droplets of molecular motors, and confirmed that motors with longer OEG chains have higher $T_c$, and $Z$ isomers exhibit higher $T_c$ than their corresponding $E$ isomers.

## The rotary motion of molecular motors drives reversible phase separation

The rotation of **2MOEG4** in assemblies was monitored by UV-Vis spectroscopy. In contrast to the rotation of **2MOEG4** as monomers, the rotation of **2MOEG4** in assemblies was accompanied by a reversible phase separation process at RT. Upon irradiation, the absorption band at 300–385 nm of $Z_S$-**2MOEG4** exhibits a decrease, accompanied by the emergence of a new band at 386–485 nm, indicating the formation of $E_M$-**2MOEG4** (Fig. 4a). An isosbestic point was observed at 386 nm was observed during the first 200 s of irradiation, which then disappeared with increasing full-spectrum absorption due to scattering from the phase separation of $E_M$-**2MOEG4**. The exponential curve of the absorption at 425 nm as the function of time until 200 s indicates photoisomerization of $Z_S$-**2MOEG4** to $E_M$-**2MOEG4** (Fig. 4b). After 200 s, the increase of the absorption at 500 nm is indicative of

LLPS. The phase separation of $E_M$-**2MOEG4** was further confirmed by DLS measurements. The hydrodynamic diameter of the assemblies increased from 7 nm to 490 nm after irradiation (Supplementary Fig. 9). The subsequent incubation of the sample in the dark for 50 min resulted in the THI of $E_M$-**2MOEG4** to $E_S$-**2MOEG4**, accompanied by phase separation, as indicated by the reduction in absorption at 425 nm and an increase in absorption at 500 nm (Fig. 4c). The isomerization of motors during LLPS can be approximated by subtracting the absorbance at 425 nm from the absorbance at 500 nm (Abs@425 − 500 nm). Abs@425−500 nm demonstrates an exponential decline over time, which is indicative of the THI$_1$ process (Fig. 4d). The enhanced scattering may be attributed to the increased size of the assembly during the THI$_1$ process, as shown by DLS data, which indicate that the hydrodynamic diameter of the assemblies grew to 890 nm (Supplementary Fig. 9). A subsequent irradiation of $E_S$-**2MOEG4** resulted in a slight increase in the absorption band at 425 nm, accompanied by a decrease in the full spectrum absorption band (Fig. 4e). The absorption at 500 nm showed a notable decline, whereas the Abs@425−500 nm increased exponentially (Fig. 4f). This implies that the transformation of $E_S$-**2MOEG4** to $Z_M$-**2MOEG4** is concomitant with the dissolution of droplets. Finally, the band at 386−485 nm disappeared with an increase of absorption band at 300−385 nm when the sample was kept in the dark, indicating the THI of $Z_M$-**2MOEG4** to $Z_S$-**2MOEG4** (Fig. 4g). An isosbestic point at 386 nm was observed during THI$_2$ (Fig. 4g) and the absorption at 500 nm was maintained at zero in this process, thereby confirming the homogeneity of the solution (Fig. 4h). As a control, the rotation of $E_S$-**2MOEG4** assemblies was also studied in water at lower temperatures. At a temperature below the $T_c$ of four isomers, no phase separation behavior was observed upon the rotation of **2MOEG4** (Supplementary Fig. 3). The same behavior was also found for **2MOEG6** (Supplementary Fig. 4). These results indicate that the rotation-driven reversible LLPS is caused by the successively different $T_c$ of the isomers. The $T_c$ of $Z$ isomers is higher than the operating temperature of the system, while the $T_c$ of $E$ isomers is lower than that, which results in the reversible LLPS during the rotation of the molecular motor.

To investigate the in situ generation of droplets, the rotation-driven phase separation was monitored using CLSM. The internal structure changes of assemblies were examined before and after phase separation using cryo-TEM. After photoinduced isomerization, the solution of $Z_S$-**2MOEG4** supramolecular assemblies (Fig. 4j) underwent LLPS to form droplets (Fig. 4l). The worm-like micelles of $Z_S$-**2MOEG4** (Fig. 4i) were transformed into tightly aggregated micelles of $E_M$-**2MOEG4** as a consequence of phase separation (Fig. 4k). During the THI of $E_M$-**2MOEG4** to $E_S$-**2MOEG4**, the assembled structure remains relatively unchanged (Fig. 4o), while the droplet size exhibits a slight increase (Fig. 4p). Upon subsequent irradiation, the aggregated micelles of $E_S$-**2MOEG4** transformed into dispersed worm-like micelles of $Z_M$-**2MOEG4** (Fig. 4m), and the droplets dissolved (Fig. 4n). Ultimately, keeping the sample in the dark led to the THI of $Z_M$-**2MOEG4** to $Z_S$-**2MOEG4**, occurring in a homogeneous solution of supramolecular assemblies without droplets (Supplementary Fig. 15). The combined CLSM and cryo-TEM study indicate that the $E_M$-**2MOEG4** and $E_S$-**2MOEG4** isomers exhibit phase separation at RT, whereas the $Z_M$-**2MOEG4** and $Z_S$-**2MOEG4** isomers do not. Consequently, the reversible LLPS is achieved during the rotation of molecular motors, in accordance with the findings of UV-Vis spectroscopy.

## Rotational speed of the molecular motor in supramolecular assemblies and droplets

To quantify the rotational speed of molecular motors in assemblies and droplets, we performed Eyring analysis on the THI processes from metastable to stable states and determined the energy landscape for motor rotation (Fig. 5, see Methods section for details). The Eyring analysis of the THI in water (assembly states) reveals an energy barrier

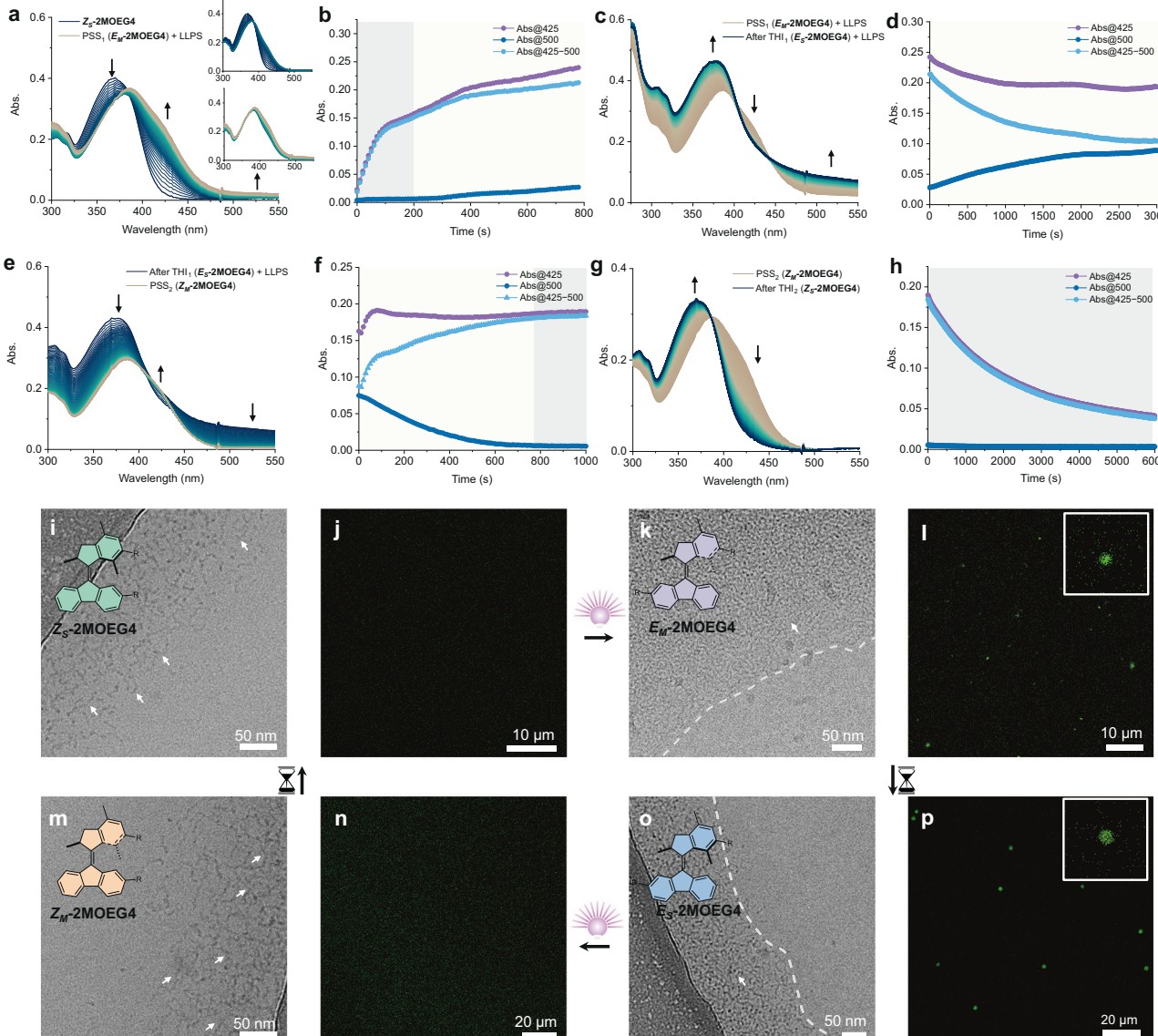

**Fig. 4 | Molecular motor-driven reversible LLPS. a** UV-Vis absorption spectra of $Z_S$-**2MOEG4** (50 µM, $H_2O$, RT) upon 365 nm light irradiation for 13 min to PSS$_1$ getting $E_M$-**2MOEG4**, accompanied with phase separation. **b** Corresponding absorption at 425 nm and 500 nm. Gray shading: isomerization; pale yellow shading: isomerization and liquid–liquid phase separation (LLPS). **c** UV-Vis absorption spectra of $E_M$-**2MOEG4** while keeping in the dark for 50 min to process THI$_1$ and reach $E_S$-**2MOEG4** with phase separation. **d** Corresponding absorption at 425 nm and 500 nm. **e** UV-Vis absorption spectra of $E_S$-**2MOEG4** upon subsequent irradiation with 365 nm light for 17 min to PSS$_2$ getting $Z_M$-**2MOEG4** with a gradual disappearance of phase separation. **f** Corresponding absorption at 425 nm and 500 nm. **g** UV-Vis absorption spectra of $Z_M$-**2MOEG4** while keeping in the dark for 100 min to process THI$_2$ and recover $Z_S$-**2MOEG4**. **h** Corresponding absorption at 425 nm and 500 nm. PSS: photostationary state; THI: thermal helix inversion. Cryo-TEM (**i**, **k**, **m**, and **o**) and CLSM images (**j**, **l**, **n**, and **p**) of an aqueous solution of (**i** and **j**) $Z_S$-**2MOEG4** assemblies, upon 365 nm light irradiation at RT to get (**k** and **l**) $E_M$-**2MOEG4**, then keeping in the dark to reach (**o** and **p**) $E_S$-**2MOEG4**, followed by subsequent irradiating with 365 nm light to get (**m** and **n**) $Z_M$-**2MOEG4**. Arrow: worm-like micelles, dash line: the boundary of tightly aggregated worm-like micelles and blank, bulb: the irradiation of light, hourglass: the passage of time.

($\Delta^{\ddagger}G°$) of 87.8 kJ mol$^{-1}$ from $E_M$-**2MOEG6** to $E_S$-**2MOEG6** (Fig. 5b), which is higher than that of 84.3 kJ mol$^{-1}$ in the monomeric state (Fig. 5a). $\Delta^{\ddagger}G°$ from $Z_M$-**2MOEG6** to $Z_S$-**2MOEG6** is 84.4 kJ mol$^{-1}$ for the monomer and 88.7 kJ mol$^{-1}$ in the assembly (Supplementary Figs. 16 and 17). Accordingly, the half-lifetime ($t_{1/2}$) of $E_M$-**2MOEG6** is 2.0 min as a monomer and 8.3 min in assemblies, while the $t_{1/2}$ of $Z_M$-**2MOEG6** is 2.1 min as a monomer and 12.1 min in assemblies at 20 °C (Table S1). $\Delta^{\ddagger}G°$ from $E_M$-**2MOEG4** to $E_S$-**2MOEG4** and $Z_M$-**2MOEG4** to $Z_S$-**2MOEG4** are 87.9 and 89.0 kJ mol$^{-1}$, respectively, which are also higher than the 84.0 and 84.2 kJ mol$^{-1}$ in the monomeric states (Supplementary Figs. 18-21). The $t_{1/2}$ of $E_M$-**2MOEG4** is 1.7 min as a monomer, while it is 8.7 min in assemblies, whereas the $t_{1/2}$ of $Z_M$-**2MOEG4** is 1.9 min as a monomer and 13.5 min in assemblies at 20 °C (Table S1). The slower

speed of motors in assemblies can be attributed to the tight packing of neighboring molecules, which restrict the rotation[75]. The energy barrier for two THI processes after phase separation is 87.5 and 88.1 kJ mol$^{-1}$ (Fig. 5c), and the half-lifetimes of $E_M$-**2MOEG4** and $Z_M$-**2MOEG4** are 7.4 min and 9.3 min at 20 °C, respectively, which are comparable to those in the assemblies before phase separation (Supplementary Figs. 22 and 23). The energy landscape of monomers and assemblies is shown in Fig. 5c. Overall, the rotational speeds of **2MOEG4** and **2MOEG6** monomers are nearly identical, but both are slower in supramolecular assemblies compared to their monomeric state. The rotational speed of **2MOEG4** remains largely unchanged after phase separation, indicating the good fluidity of the phase-separated droplets.

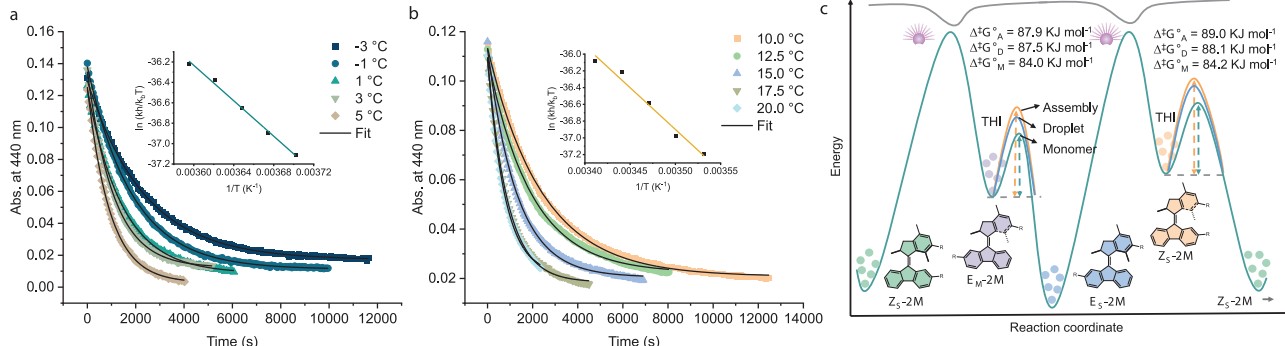

**Fig. 5 | Energy landscape for the rotary motion of molecular motor.**
**a** Absorption at 440 nm as a function of time during the THI of $E_M$-**2MOEG6** (50 µM) in MeOH (as monomers) at different temperatures. **b** Absorption at 440 nm as a function of time during the THI of assemblies of $E_M$-**2MOEG6** (50 µM) in water at different temperatures. Inset: Eyring plots with a linear fit. **c** Energy landscape diagram for the rotary motion of molecular motors as monomers, in assemblies, and droplets, respectively. The illustration conceptually shows the energy barriers differences. THI: thermal helix inversion, bulb: the irradiation of light.

In summary, we have developed a molecular motor-driven LLPS of supramolecular assemblies, enabling orthogonal control by light and heat. Supramolecular assemblies of molecular motors exhibit different morphologies, deviating from spherical micelles to worm-like micelles and fibers, depending on the molecular structures and growth kinetics. The droplets formed through phase separation exhibit excellent fluidity, as demonstrated by the rapid recovery of fluorescence from the labeling dye and the uninhibited rotational speed of molecular motors. The critical phase separation temperatures are found to be closely correlated with the nature of hydrophilic ethylene glycol chain units and the configurations of molecular motors. The large geometrical differences between the isomers result in distinct critical phase separation temperatures, ranging from 18 to 52 °C. The reversible phase separation can be regulated by modifying the temperature, utilizing different critical phase separation temperatures. Benefiting from the multiple out-of-equilibrium states of molecular motors and their distinct critical phase separation temperatures, in situ reversible phase separation is realized during the unidirectional rotation of molecular motors. Achieving in situ control over the LLPS of supramolecular assemblies will facilitate the advancement of out-of-equilibrium biomedical materials with responsive capture and release functions.

## Methods
General information and the synthesis of molecules are outlined in Supplementary Information. Supramolecular assemblies of molecular motors were prepared by dissolving them in Milli-Q water, followed by 1 min of ultrasonic treatment and stabilization at room temperature for 1 h.

### Ultraviolet–visible (UV-Vis) spectroscopy study
UV-Vis spectra were recorded on a Hewlett-Packard HP 8543 spectrometer in a quartz cuvette with a 1 cm path length. Irradiation of samples was carried out in situ using an LED light M365FP1 (5.29 mW model) positioned at a distance of 2 cm from the sample. Transmittance measurements were recorded on a Jasco V-750 spectrometer in a quartz cuvette with a 1 cm path length. The transmittance of samples was monitored at the wavelength of 500 nm. Temperature-dependent transmittance measurements were performed with a heating rate of 1 °C/min. Phase transition temperatures measured from transmittance were used to plot the phase diagram.

### Nuclear magnetic resonance (NMR) spectroscopy study
NMR studies on the rotation of molecular motors were performed on Varian Unity Plus (¹H: 500 MHz, ¹³C: 125 MHz) spectrometer. Irradiation of samples was carried out in situ using an LED light M365FP1 (5.29 mW model) equipped with optical fiber placed in the sample. NMR spectra were recorded on Varian AMX400 (¹H: 400 MHz, ¹³C: 101 MHz) and Varian Unity Plus (¹H: 500 MHz, ¹³C: 125 MHz) spectrometers. To stabilize metastable isomers during NMR analysis, measurements of metastable isomers were conducted at low temperatures (−20 °C).

### Cryogenic transmission electron microscopy (cryo-TEM) study
A water solution of $Z_S$-**2MOEG4** (0.5 mg/mL) was treated with ultrasound to be dispersed and kept in the dark at RT for 1 h (Fig. 4i). The sample was first irradiated with 365 nm light for 10 min to get $E_M$-**2MOEG4** in a quartz cuvette with 1 mm path length (Fig. 4k). The obtained sample was kept in the dart at RT for 60 min to reach $E_S$-**2MOEG4** state (Fig. 4o). The sample was subsequent irradiating with 365 nm light for 10 min to get $Z_M$-**2MOEG4** (Fig. 4m). Finally, the sample was kept in the dark at 25 °C for 60 min to recover $Z_S$-**2MOEG4**. In each state, a few microliters of sample solution were placed on holey carbon-coated copper grids (Quantifoil 3.5/1, Quantifoil Micro Tools, Jena, Germany). Grids with samples were vitrified in liquid ethane (Vitrobot, FEI, Eindhoven, The Netherlands) and transferred to a FEI Tecnai T20 cryo-electron microscope operating at 200 keV. Images were recorded under low-dose conditions with a slow-scan CCD camera. $Z_S$-**2MOEG6** and aged $Z_S$-**2MOEG4** solutions were measured using the same method.

### Dynamic light scattering (DLS) measurements
The water solution of $Z_S$-**2MOEG4** (0.5 mg/mL) was treated with ultrasound for 1 min to be dispersed and stabilized at RT for 1 h. The aqueous sample solution was placed in a plastic cuvette. DLS measurements were conducted on Zetasizer Ultra equipment with a fluorescence filter using He-Ne laser (633 nm). All the setups were calibrated before measurements. For each temperature, five measurements were performed after equilibrium for 300 s. Data were analyzed in the ZS XPLORER software, assuming a refractivity index of 1.56. The size of assemblies was determined following the number distribution. Temperature-dependent measurements were taken every 5 °C, and the sample was held at each temperature for 5 min to reach equilibrium. The measurements for the aging effect on the assemblies were conducted at RT.

### Confocal laser scanning microscopy (CLSM) study
The water solution of $Z_S$-**2MOEG4** (0.5 mg/mL) was treated with ultrasound for 1 min to be dispersed and stabilized at RT for 1 h. The light irradiation procedure and thermal helix inversion were the same as for the cryo-TEM study. Samples were dyed with 0.5 µM of Nile Red. 9.2 µL of each sample was loaded into a 120 µm thick sample chamber consisting of two coverslips and an imaging spacer in the center (Grace Bio-Labs SecureSeal Imaging Spacer, diameter: 9 mm) and measured via

Leica TCS SP8 equipped with a 63 × 1.2 (water immersion) numerical aperture objective. A laser of 552 nm was used as the excited light source, and the emission was recorded at the wavelength range of 580–750 nm.

Temperature-controlled CLSM studies were performed using a VaHeat (Interherence) system. 2 µL of sample solution was loaded into a glass capillary (0.2 × 3 × 15 mm) and sealed for temperature-variable measurements. The glass capillary was placed on the VaHeat microscopy-compatible stage, and temperature was controlled in situ on Leica TCS SP8 equipped with a 40 × 1 numerical aperture objective.

### Fluorescence recovery after photobleaching (FRAP)

The water solution of $Z_S$-2MOEG4 (0.5 mg/mL) was dyed with 0.5 µM of Nile Red. Two images were taken before bleaching at an imaging power of 1.0%, using a laser of 552 nm as the excitation light source, recorded at the wavelength range of 580–750 nm. Subsequently, a defined region of interest (ROI; 6 to 8 µm in diameter) was photobleached for 10 cycles (1.29 s per cycle) at 100% of the power with the 552 nm laser. After bleaching, the following images were automatically captured every 1.29 s at 1.0% power to record the fluorescence recovery.

Fluorescence intensity within the bleached ROI, and a background region was quantified using the built-in software LAS X. Intensities were background-subtracted and normalized to pre-bleach values.

Normalized recovery curves were fitted to a single-exponential model:

$$F(t) = F_\infty (1 - e^{-t/\tau}) \tag{1}$$

where:

$F(t)$ is the normalized fluorescence at time t.

$F_\infty$ is the plateau fluorescence after full recovery.

$\tau$ is the characteristic recovery time constant.

The half-life time of recovery ($t_{1/2\ \mathrm{FRAP}}$) was calculated from the time constant $\tau$ as:

$$t_{1/2\mathrm{FRAP}} = \tau \times \ln(2) \tag{2}$$

### Eyring analysis on the thermal helix inversion of molecular motors

The kinetic files of the THI of molecular motors as monomers, in assemblies, and in droplets were recorded on a Hewlett-Packard HP 8543 spectrometer at different temperatures. Kinetic data were fitted with exponential functions to obtain rate constants ($k$) at different temperatures[76]. The values of ln($kh/K_BT$) were plotted as a function of $1/T$, and linear fitting was performed to extrapolate the rate constant at 273 K. Standard Gibbs free energy of activation $\Delta^{\ddagger}G^\circ$ for THI was determined at 273 K using the Eyring equation.

$$\Delta^{\ddagger}G^\circ = RT\left(\ln\frac{k_B}{h} - \ln\frac{k}{T}\right) \tag{3}$$

where:

$R$ is universal gas constant (8.314 J mol$^{-1}$ K$^{-1}$).

$T$ is the absolute temperature ($K$).

$k_B$ is Boltzmann constant (1.3806 × 10$^{-23}$ J K$^{-1}$).

$h$ is Planck constant (6.6261 × 10$^{-34}$ J s).

$k$ is the rate constant of the reaction (s$^{-1}$).

The half-life ($t_{1/2}$) of the metastable isomers was determined at 273 K using the following equation:

$$t_{1/2} = \frac{\ln 2}{k} \tag{4}$$

## Data availability

The data generated in this study are provided in the Supplementary Information and Source Data file. Data are available from the corresponding author on request. Source data are provided with this paper.

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

## Acknowledgements

The author would like to express their gratitude to Prof. E. W. (Bert) Meijer for his advice and support, Dr. Yu Zhao for DFT calculations, Barbara Malheiros for her guidance in CLSM and Dr. Huaizhou Yang for help with formatting revisions. This work was supported by the Netherlands Organization for Scientific Research (B.L.F.), the Royal Netherlands Academy of Arts and Sciences (B.L.F.), and the Dutch Ministry of Education, Culture, and Science (Gravitation Program 024.001.035 to B.L.F.).

## Author contributions

B.L.F. supervised the project. F.X. and B.L.F. conceived the project. F.X. performed most experiments and drafted the manuscript. M.O. co-preformed the DLS study, prepared illustration schemes, and helped to write the manuscript. Y.F. conducted exploratory fluorescence spectroscopy measurements. M.A.C.S. performed cryo-TEM measurements. All authors discussed the results and revised the manuscript.

## Competing interests

The authors declare no competing interests.
