## [Transparent Peer Review file · Nature Communications]

Molecular Motor-driven Reversible Liquid-liquid Phase Separation of Supramolecular Assemblies

Corresponding Author: Professor Ben Feringa

Version 0:

Reviewer comments:

Reviewer #1

(Remarks to the Author)

Prof. Feringa and coworkers have demonstrated with a molecular motor-based liquid-liquid phase separation (LLPS) system with full controls with UV-light and heating processes. The authors have highlighted that only several examples of synthetic molecular structures were demonstrated as LLPS systems, more importantly, they attempted to control the structural difference among the isomers of molecular motors to induce variations in hydrophobicity and hydrogen bonding for LLPS controls, though these points remained unaddressed. By tuning the length of hydrophilic motifs of the bolaamphiphilic design of molecular motors, the molecular motor amphiphiles can be systematically adjusted for possible LLPS states at ~25 °C. The detailed investigations with cryo-TEM and confocal microscopy have demonstrated the LLPS of supramolecular assemblies of motor amphiphiles. However, some minor concerns should be addressed before further considering for publishing in the Nature Communications.

- 1) The authors should show with experimental evidence on the role of urea motif as part of the molecular design, such as NMR. Besides, the authors should mention how the urea motif is packed in the nanoassemblies to afford significant supramolecular structural difference in aqueous media.
- 2) Aging studies of supramolecular nanostructures were provided, but correlations to LLPS of the motor amphiphiles should also be addressed?
- 3) Figure 2a, the spectrum (after 360° rotation) of Zs-2MOEG4 shows three set of peaks with chemical shift at 7.42, 7.05, and 7.00 ppm, which are absence before irradiation. The authors might comment with reasons, such as photodegradation or other causes. It is suggested that full NMR spectra of the photothermal isomerization processes should be provided in the supporting information.
- 4) To the reviewer understanding, Zs-2MOEG4 should be obtained from synthetic pathway directly. What is the reason for the authors naming the motor amphiphile as bola-amphiphile with both hydrophilic motifs on the same side of the motor core?
- 5) An extra space was found in Figure 2e. After TH12 (Zs-2MOEG4).
- 6) What kind of morphological measurement setting has been used for determining the size distribution in DLS for Aged Zs-2MOEG4 to observe hydrodynamic diameter of 150 nm? Please specify in the supporting information.
- 7) The schematic illustrations of supramolecular assemblies for Zs-2MOEG are seemingly organized structures, how can the reader correlate the Figure 1 to the worm-like micelle as observed in Figure 2f and 2g?
- 8) Should the nanostructures inherit supramolecular helicities in aqueous media? The correlated CD studies should be provided.
- 9) LLPS investigations in Figure 3 have been well demonstrated in term of morphological transformations. In addition to DLS and cryo-TEM, any packing structural investigations can be performed such as small-angle X-ray scattering in solution?
- 10) The cryo-TEM images have been well recorded the correlate nanostructures with the microscopic phase separation with doped Nile Red in confocal microscopy. The authors named the boundary of tightly aligned worm-like micelles in Figure 4l and 4p. What do they mean on the term of "aligned" and aligned to what subject?
- 11) Figure 5c illustrated the energy landscape of the three states monomer, assembly, and droplet. But the measuring conditions for monomers were in methanol, is it proper to keep in same energy landscape with neglecting the contributions from surrounding environment?
- 12) As mentioned in the introduction, the authors are suggested to discuss on how structural difference among the isomers of molecular motors induce variations in hydrophobicity and hydrogen bonding for LLPS controls.

Reviewer #2

(Remarks to the Author)

- What are the noteworthy results?

The authors achieve the reversible switching of liquid-liquid phase separation within a molecular assembly. Whilst well presented, the publication lacks in novelty.

- Will the work be of significance to the field and related fields? How does it compare to the established literature? If the work is not original, please provide relevant references.

The authors use a modification of their widely reported crowded alkene motor to switch between two distinct states of an assembly. However, in this case the directional motion provided by the motor is inconsequential, it is only the differences in geometrical structures provided by the E and Z forms that matter in regards to the properties of the system. Indeed, this point is illustrated in a related recent study reported in Nature Materials not cited by the authors (Deng et al. DNA photofluids show life-like motion. Nat. Mater. (2025). <https://doi.org/10.1038/s41563-025-02202-0> -) where similar results are achieved – the reversible formation and dissolution of droplets with light as a stimulus using a simple azo benzene switch.

- Does the work support the conclusions and claims, or is additional evidence needed?

The work does support the hypothesis that the large change in the geometric shape arising from light driven isomerisation of the alkene motor leads to the formation and dissolution of droplets. I feel that more emphasis needs to be made on why using a directional motor to do this (rather than a simple switch) is important. What does the motor do that isn't possible with an azobenzene? What is the advantage of having directionality in a system that doesn't intrinsically need it?

- Are there any flaws in the data analysis, interpretation and conclusions? Do these prohibit publication or require revision?

The supporting information provided looks sufficient. However as mentioned the light driven isomerization and photophysical properties of these systems has been thoroughly studied by the group and others, so the presented results are hardly groundbreaking.

- Is the methodology sound? Does the work meet the expected standards in your field?

The methodology is sound, the system uses (as mentioned above) the well-studied crowded alkene motor as its base.

- Is there enough detail provided in the methods for the work to be reproduced?

The experiments in supporting information appear reproducible.

Reviewer #3

(Remarks to the Author)

This manuscript reported the preparation of molecular motor-based supramolecular assemblies for light and heat responsive LLPS. The molecular motors with specific unidirectional four-state rotation allow controlled supramolecular assembly and reversible LLPS based on geometric dependent LCST phase behavior. The perspective is attractive and present significant inspiration for supramolecular biofunctional materials. However, the lack of logical compactness and experimental rationality request optimization, which may merit publication after major revisions, the following comments are suggested to be considered:

1. The introduction presented is vague and the novelty is unclear. It seems like the authors do spread a new platform for LLPS based on molecular motors. However, they make insufficient understanding of the research background on LLPS. For instance, small molecular amphiphiles such as photoisomerization of azobenzene-derivatives have been intensively employed to stimuli-responsive coacervate assembly (Sci. Adv., 2021, 7, eabf9000, Nat. Chem., 2024, 16, 158). In this perspective, the novelty of this work may not be acknowledged by the readers unless making comprehensive review. The emphasis can focus more on features of molecular motors and their potential in developing biofunctional materials. Finally, I strongly recommended the author reorganize the introduction.

2. What is the transition ratio of EM to ES? Despite of extensive studies in molecular motors, the authors should provide basic definition of each rotation state including their differences in stereogenic orientation, physical parameters and transition mechanisms during photoisomerization and THI.

3. Why the authors chose second-generation molecular motor amphiphiles?

4. The scheme representing supramolecular assemblies in Figure 1 is confusing and misleading. Are they twisted nanofibrils or homogeneous aqueous solutions? Please replace identical expressions and avoid misleading, or the authors may provide CD spectra or AFM as evidence of helical assembly.

5. According to monomer rotation studies, NMR and UV-Vis are carried out under distinct temperature treatment, this is confusing. Is the THI steps primary a result of temperature or time-dependent dynamic relaxation?

6. The Cryo-TEM image of figure 2f is poor in quality and request optimization with higher magnification. Besides, please supplement experimental details in developing supramolecular assemblies.

7. According to thermal-responsive LLPS, how does stable E isomers be obtained? Why only stable E/Z isomers are discussed instead of metastable states? Additionally, please compare fluidity of ES-2MOEG4, ZS-2MOEG4, ES-2MOEG6 and ZS-2MOEG6.

8. The discussion of LLPS during rotary motion is controversial. For example, Figure 4a makes no significant trend of full-spectrum absorption and more representative tests are request. Figure 4i reflected negligible LLPS while the author claimed "the solution of ZS-2MOEG4 supramolecular assemblies underwent LLPS to form droplets (Fig. 4i and k)". Additionally, the

conclusion at the end of first paragraph stated that reversible LLPS can be induced under $T_c(Z) < T < T_c(E)$, which is contradictory to general trend $T_c(E) < T_c(Z)$. Please clarify the above discrepancy and revise your article carefully.

9. Why both energy barrier and half-lifetimes of assemblies slightly decreased after phase separation.

10. Whether two-modal control can be orthogonal coupled for LLPS regulation?

Version 1:

Reviewer comments:

Reviewer #1

(Remarks to the Author)

The authors have fully addressed all questions and comments requested by the reviewer. The reviewer is now supporting the current version of manuscript for publishing in the Nat. Commun.

Reviewer #2

(Remarks to the Author)

The authors have made extensive changes to the manuscript and supporting information. I feel these changes strengthen the research and really help highlight the novelty of the work.

As such I recommend that the article be accepted for publication.

Reviewer #3

(Remarks to the Author)

The revisions have fully addressed my concerns. The authors demonstrated commendable diligence and scholarly rigor in implementing suggested improvements. I therefore recommend acceptance of the manuscript in its current form.

Detailed response to reviewers

We appreciate the positive response and scholarly suggestions by the reviewers. A detailed response to all points raised is presented in the following.

Reviewer #1 (Remarks to the Author):

Prof. Feringa and coworkers have demonstrated with a molecular motor-based liquid-liquid phase separation (LLPS) system with full controls with UV-light and heating processes. The authors have highlighted that only several examples of synthetic molecular structures were demonstrated as LLPS systems, more importantly, they attempted to control the structural difference among the isomers of molecular motors to induce variations in hydrophobicity and hydrogen bonding for LLPS controls, though these points remained unaddressed. By tuning the length of hydrophilic motifs of the bolaamphiphilic design of molecular motors, the molecular motor amphiphiles can be systematically adjusted for possible LLPS states at ~25 °C. The detailed investigations with cryo-TEM and confocal microscopy have demonstrated the LLPS of supramolecular assemblies of motor amphiphiles. However, some minor concerns should be addressed before further considering for publishing in the Nature Communications.

We sincerely thank the reviewer for their thorough evaluation and insightful comments. Below we provide our detailed responses and indicate the revisions made to the manuscript and Supplementary Information accordingly.

1) The authors should show with experimental evidence on the role of urea motif as part of the molecular design, such as NMR. Besides, the authors should mention how the urea motif is packed in the nanoassemblies to afford significant supramolecular structural difference in aqueous media.

We thank the reviewer for this important suggestion. We have measured the ¹H NMR spectra of Z_S-2MEG4 in D₂O. The spectra display significant broadening of the proton resonances at 4.0–3.0 ppm (hexaethylene glycol) and 1.5–1.0 ppm (alkyl chain), indicating aggregation behavior in aqueous solution (Fig. R1). Similar broadening has been reported for hexaethylene glycol-substituted molecular motor amphiphiles (*J. Am. Chem. Soc.* **2024**, 146, 23, 15843–15849). Therefore, due to the aggregation and resulting broad signals, direct observation of the urea motif by NMR in aqueous solution remains challenging.

Fig. R1 ^1H NMR of Z_5 -2MEG4 in D_2O (400 MHz, 2 mg/ml).

The cryo-TEM and DLS study indicate a significant supramolecular structural difference of fresh and aged assembly of Z_5 -2MEG4. The fresh assembly shows worm-like micelles, whereas the aged sample transforms into straight fibers with micrometer-scale lengths. To further investigate how the urea motifs are packed within the nanoassemblies over time, FTIR measurements were performed on samples before and after aging for 1 week (Fig. R2). The fresh sample of Z_5 -2MOEG4 (worm-like micelles) exhibited a N–H stretching centered at 3358 cm^{-1} and a C=O stretching at 1641 cm^{-1} , attributed to moderate-strength hydrogen-bonded N–H in urea moieties, both characteristic of moderately hydrogen-bonded urea moieties. In contrast, the aged sample of Z_5 -2MOEG4 (straight fibers) showed an N–H stretching band at 3338 cm^{-1} and a C=O stretching band at 1625 cm^{-1} , corresponding to approximately 20 cm^{-1} red-shifts compared to the fresh assembly. These shifts suggest enhanced urea hydrogen bonding upon aging, supporting the transformation from worm-like micelles to micrometer-scale straight fibers.

We have included the FTIR data and discussion in the revised manuscript (Fig.2j; Page 5, line 139-143).

Fig. R2 (Fig. 2j) FTIR spectra of the Z_5 -2MOEG4 assembly and the same sample after aging in water for one week. Samples were analyzed after drying from aqueous solution.

2) Aging studies of supramolecular nanostructures were provided, but correlations to LLPS of the motor amphiphiles should also be addressed?

We appreciate the reviewer's suggestion. We have investigated the LLPS behavior of aged supramolecular assemblies using temperature-dependent transmittance measurements and DLS study. At the same concentration, the critical temperature (T_c) of the aged **Z_S-2MOEG4** assembly solution was found to decreased by 4 °C (Fig. R3). Cryo-TEM measurements reveal that the length of worm-like micelles grows from ~20 nm in the fresh state to several micrometers after aging. The decrease of T_c may be attributed to the increased length of supramolecular polymers upon aging. This observation is consistent with trends seen in covalent polymers exhibiting LCST behavior, where increasing molecular weight leads to a lower LCST (*Chem. Soc. Rev.*, **2013**, 42, 7468).

DLS studies revealed that the hydrodynamic diameter of the droplets formed by aged **Z_S-2MOEG4** assemblies is around 400 nm upon phase separation (Fig. R4). Moreover, once the temperature decreases below T_c , the assemblies reverted to their original size.

We have included these results and discussion in the manuscript (Page 6, line 167-168 and 182-183) and figures in the Supplementary Information (Supplementary Figs. 6 and 10).

Fig. R3 (Supplementary Fig. 6) Transmittance of aqueous solutions of fresh and aged **Z_S-2MOEG4** (0.3 mg/mL) upon heating.

Fig. R4 (Supplementary Fig. 10) Time-dependent DLS data of aged **Z_S-2MOEG4** assembly in water.

3) Figure 2a, the spectrum (after 360° rotation) of Zs-2MOEG4 shows three set of peaks with chemical shift at 7.42, 7.05, and 7.00 ppm, which are absence before irradiation. The authors

might comment with reasons, such as photodegradation or other causes. It is suggested that full NMR spectra of the photothermal isomerization processes should be provided in the supporting information.

We thank the reviewer for this observation. The peaks mentioned correspond to different isomers of the motor rather than degradation. This is due to the photoisomerization process that although it is highly efficient (~60 to ~80% conversion), is not quantitative. Thus, a fraction of the unreacted isomer is present during the next step. The final mixture after a rotation cycle contains **E_S-2MOEG4**. We have provided the full ¹H NMR spectra (Supplementary Fig. 38) of the photoisomerization and THI process in the revised Supplementary Information for clarity and specify the ratios of the photochemical steps in the manuscript (Page 3, line 93-100).

4) To the reviewer understanding, Zs-2MOEG4 should be obtained from synthetic pathway directly. What is the reason for the authors naming the motor amphiphile as bola-amphiphile with both hydrophilic motifs on the same side of the motor core?

Yes, **Zs-2MOEG4** was obtained by the synthetic route. Thanks to the reviewer for correcting the definition. When the motor is in the Z configuration, the two hydrophilic chains do lie on the same side of the motor core, but the two hydrophilic chains are not oriented in the same direction. After isomerization to the E-isomer, the two hydrophilic chains are located on both sides of the motor core. Therefore, we call this molecule a bola-amphiphile. To avoid misunderstanding, we have changed this term to the broader term "amphiphile".

5) An extra space was found in Figure 2e. After THI2 (Zs-2MOEG4).

We thank the reviewer for carefully identifying this typographical error. We have corrected the figure by removing the extra space (Fig. 2e).

6) What kind of morphological measurement setting has been used for determining the size distribution in DLS for Aged Zs-2MOEG4 to observe hydrodynamic diameter of 150 nm? Please specify in the supporting information.

We appreciate this important point. As standard DLS assumes spherical particle morphology, there are inherent limitations when analyzing elongated structures such as fibers. The reported hydrodynamic diameter reflects an equivalent scattering radius based on Brownian motion, rather than the actual length of supramolecular fibers. The primary purpose of the DLS measurement was to qualitatively compare the assemblies before and after aging, rather than to precisely define fiber dimensions. The size and morphology of aged **Z_S-2MOEG4** were characterized by Cryo-TEM studies. We have clarified these points and provided detailed DLS measurement conditions in the revised Supplementary Information (Page10).

7) The schematic illustrations of supramolecular assemblies for Zs-2MOEG are seemingly organized structures, how can the reader correlate the Figure 1 to the worm-like micelle as observed in Figure 2f and 2g?

We appreciate the reviewer's observation. To better reflect the actual morphology, we have revised the schematic illustration in Fig. 1 so that it more closely resembles the worm-like micelle structures observed in the Cryo-TEM images.

8) Should the nanostructures inherit supramolecular helicities in aqueous media? The correlated CD studies should be provided.

We thank the reviewer for this valuable comment. As the molecular motor used here is a racemic mixture, the supramolecular assemblies are not expected to display chiral order or helicity, which was confirmed by Cryo-TEM analysis. Nonetheless, to directly address the reviewer's suggestion, we performed CD measurements on **Z_S-2MOEG4** assemblies. The CD spectra showed no signal, confirming the absence of supramolecular chirality (Fig. R5). We have included the CD data and corresponding discussion in the revised Supplementary Information (Page 20; Supplementary Fig. 27).

Fig. R5 (Supplementary Fig. 27) CD and UV-vis absorption spectra of **Z_S-2MEG4** assembly (1mg/mL) in water in a cuvette with 1mm light pathlength.

9) LLPS investigations in Figure 3 have been well demonstrated in term of morphological transformations. In addition to DLS and cryo-TEM, any packing structural investigations can be performed such as small-angle X-ray scattering in solution?

We thank the reviewer for this suggestion. In our previous work (*J. Am. Chem. Soc.* **2024**, 146, 23, 15843–15849), we have attempted to analyze the packing structures of molecular motor amphiphiles in water using SAXS. However, due to the less ordered packing nature of these supramolecular assemblies, SAXS provided structural information of overall morphology, which remains consistent with the Cryo-TEM observations. Therefore, we believe that in our system, SAXS and other X-ray scattering techniques will not offer additional information to our Cryo-TEM and DLS studies.

10) The cryo-TEM images have been well recorded the correlate nanostructures with the microscopic phase separation with doped Nile Red in confocal microscopy. The authors named the boundary of tightly aligned worm-like micelles in Figure 4l and 4p. What do they mean on the term of “aligned” and aligned to what subject?

We thank the reviewer for this observation. We intend to describe that these micelles are aggregated together. Indeed, aligned is not the best word to describe this. We have changed it to “aggregated”.

11) Figure 5c illustrated the energy landscape of the three states monomer, assembly, and

droplet. But the measuring conditions for monomers were in methanol, is it proper to keep in same energy landscape with neglecting the contributions from surrounding environment?

We appreciate the reviewer's insightful question. The absolute energy values are influenced by solvent environments, but the qualitative energy barriers remain informative. The intent of the illustration was to conceptually describe the energy barriers differences between monomeric, assembled, and LLPS states. We have added a note in the manuscript to clarify this conceptual simplification (Page 10, line 283-284).

12) As mentioned in the introduction, the authors are suggested to discuss on how structural difference among the isomers of molecular motors induce variations in hydrophobicity and hydrogen bonding for LLPS controls.

We thank the reviewer for this insightful suggestion. To assess hydrogen bonding, we performed FTIR spectroscopy on **Z_S-2MOEG4** and **E_S-2MOEG4** assemblies. Both assemblies displayed N–H stretching centered at 3358 cm⁻¹, consistent with hydrogen-bonded urea moieties (Fig. R6). A minor redshift in the C=O stretching band from 1641 cm⁻¹ (**Z_S**) to 1637 cm⁻¹ (**E_S**) suggests slightly stronger hydrogen bonding in **E_S-2MOEG4**.

Nile Red fluorescence assay was used to probe the internal hydrophobicity of assemblies. The critical aggregation concentration (CAC) of **E_S-2MOEG4** is 2.0 μM, higher than the 0.5 μM of **Z_S-2MOEG4**, suggesting that greater hydrophobicity of **E** isomer (Fig. R6).

Together, these results demonstrate that molecular geometry influences both hydrophobicity and hydrogen bonding, which in turn modulate the LLPS behavior of the assemblies. We have now incorporated this discussion and into the revised manuscript (Page 5, line 159-164), and experiments details and supporting data Supplementary Information (Page 13, Supplementary Figs. 11 and 12).

Fig. R6 (Supplementary Fig. 11) FTIR spectra of supramolecular assemblies formed by **Z_S-2MEG4** and **E_S-2MEG4**. Samples were characterized following drying from aqueous solution.

Fig. R7 (Supplementary Fig. 12) The blueshift of Nile Red fluorescence as a function of Z_S -2MEG4 and E_S -2MEG4 concentration.

Reviewer #2 (Remarks to the Author):

- What are the noteworthy results? The authors achieve the reversible switching of liquid-liquid phase separation within a molecular assembly. Whilst well presented, the publication lacks in novelty.

We thank the reviewer for their evaluation. While we respectfully disagree with the comment, we would like to highlight the novelty of our work:

1. We developed an LLPS system using only a single, geometrically well-defined small molecule, without polymers or coacervates, which is largely unexplored.

2. The use of a molecular motor capable of sequential isomerization through four states enables unprecedented tunability in LLPS regulation, clearly distinguished from the so far studied azobenzene two-states systems.

3. The in-depth analysis correlating molecular structure with LLPS behavior of supramolecular assemblies provides deeper insight into the design principles of LLPS systems.

- Will the work be of significance to the field and related fields? How does it compare to the established literature? If the work is not original, please provide relevant references. The authors use a modification of their widely reported crowded alkene motor to switch between two distinct states of an assembly. However, in this case the directional motion provided by the motor is inconsequential, it is only the differences in geometrical structures provided by the E and Z forms that matter in regards to the properties of the system. Indeed, this point is illustrated in a related recent study reported in Nature Materials not cited by the authors (Deng et al. DNA photofluids show life-like motion. Nat. Mater. (2025). <https://doi.org/10.1038/s41563-025-02202-0>) where similar results are achieved – the reversible formation and dissolution of droplets with light as a stimulus using a simple azobenzene switch.

We thank the reviewer for highlighting this important paper. This study was published after our original submission, and we have now cited it in the revised manuscript. This very recent work highlights the field's growing interest in achieving control of LLPS using light.

However, our work has substantial differences compared to the aforementioned study:

1. In contrast to the extensively used two-state switch azobenzene, our molecular motor accesses four states and in a precisely defined sequential manner. The unique and distinctive directional motion enables multistate control of LLPS and benefits the execution of sequential and complex tasks.

2. Beyond light responsiveness, we implemented temperature-control to orthogonally modulate LLPS.

3. Rather than achieving life-like motion, our focus is on the molecular-level design principles that govern LLPS behavior, providing deeper insight into the design principles of LLPS systems.

- Does the work support the conclusions and claims, or is additional evidence needed? The work does support the hypothesis that the large change in the geometric shape arising from light driven isomerisation of the alkene motor leads to the formation and dissolution of

droplets. I feel that more emphasis needs to be made on why using a directional motor to do this (rather than a simple switch) is important. What does the motor do that isn't possible with an azobenzene? What is the advantage of having directionality in a system that doesn't intrinsically need it?

We thank the reviewer for recognizing that the data support our conclusions. We also appreciate the opportunity to clarify the differences and advantages:

1. Azobenzene is a two-state geometrical switch, while molecular motors access four states sequentially, enabling multistate, complex tasks.
2. Azobenzene inherently suffers from thermal back-isomerization, which limits long-term stability and precise control. In contrast, molecular motors offer a unique sequence of changes that do not undergo thermal back-relaxation.
3. Directional molecular motion in LLPS systems offers a platform for directional energy flow toward non-equilibrium soft matter, where the sequence of transitions, and not just the states themselves, matters.

Together with the responses to comments 1 and 2, we have incorporated these points into the revised introduction to better highlight the novelty and advantages of our work (Page 2, line 45-65).

- Are there any flaws in the data analysis, interpretation and conclusions? Do these prohibit publication or require revision? The supporting information provided looks sufficient. However as mentioned the light driven isomerization and photophysical properties of these systems has been thoroughly studied by the group and others, so the presented results are hardly groundbreaking.

We appreciate the reviewer's positive assessment that the supporting data are sufficient. As correctly noted, the photophysical properties of these molecular motors have been extensively studied in various environments by our group in the past decade; however, their behavior in LLPS systems was unknown prior to this study. Most importantly, we demonstrated that motor rotation is minimally affected by confinement within the liquid droplets, suggesting an efficient nanoscale energy conversion. In contrast, significantly hindered rotation was observed previously in confinement.

- Is the methodology sound? Does the work meet the expected standards in your field? The methodology is sound, the system uses (as mentioned above) the well-studied crowded alkene motor as its base.

We thank the reviewer for acknowledging that the methodology is sound and that the system builds on the known properties of overcrowded alkene molecular motors.

- Is there enough detail provided in the methods for the work to be reproduced? The experiments in supporting information appear reproducible.

We appreciate the reviewer's evaluation that the experiments and Supplementary Information provide sufficient detail for reproducibility.

Reviewer #3 (Remarks to the Author):

This manuscript reported the preparation of molecular motor-based supramolecular assemblies for light and heat responsive LLPS. The molecular motors with specific unidirectional four-state rotation allow controlled supramolecular assembly and reversible LLPS based on geometric dependent LCST phase behavior. The perspective is attractive and present significant inspiration for supramolecular biofunctional materials. However, the lack of logical compactness and experimental rationality request optimization, which may merit publication after major revisions, the following comments are suggested to be considered:

We thank the reviewer for the scholarly evaluation of our work and the suggestions that improved the quality of our paper.

1. The introduction presented is vague and the novelty is unclear. It seems like the authors do spread a new platform for LLPS based on molecular motors. However, they make insufficient understanding of the research background on LLPS. For instance, small molecular amphiphiles such as photoisomerization of azobenzene-derivatives have been intensively employed to stimuli-responsive coacervate assembly (Sci. Adv., 2021, 7, eabf9000, Nat. Chem., 2024, 16, 158). In this perspective, the novelty of this work may not be acknowledged by the readers unless making comprehensive review. The emphasis can focus more on features of molecular motors and their potential in developing biofunctional materials. Finally, I strongly recommended the author reorganize the introduction.

We thank the reviewer for this suggestion. While the mentioned papers were cited in the original manuscript (ref 19 and 31) we did not discussed this work in detail as part of our introduction. We now revised the manuscript incorporating a more detailed description of these previous works and the advantages that our motor-based system offers (Page 2, line 45-65).

2. What is the transition ratio of EM to ES? Despite of extensive studies in molecular motors, the authors should provide basic definition of each rotation state including their differences in stereogenic orientation, physical parameters and transition mechanisms during photoisomerization and THI.

We thank the reviewer for this suggestion. The thermal relaxation for molecular motors tend to be almost quantitative, including this study. For the photochemical steps, this is not necessarily the case. While the PSS might vary widely between motors, our system proved to have high efficiency (~60 to ~ 80%). We have now provided this discussion in the main text.

The stereogenic orientation at the chiral center varies between enantiomers and isomers and has been well documented in the literature (*Nat. Commun.* **2022**, 13, 2124). However, since a racemic mixture of the molecular motor was used in this study, the specific orientation of individual enantiomers is cancelled with their counterpart.

To define the physical parameters relevant to our study, we calculated the dipole moments of the *ZS* and *ES* isomers using DFT (Fig R8). The *ZS* isomer exhibits a dipole moment of 9.65 D, whereas the *ES* isomer has a lower value of 6.49 D. This lower dipole moment indicates reduced polarity for the *ES* isomer, consistent with the Nile Red assay results showing higher hydrophobicity for the *ES* form.

The mechanism of photoisomerization involves light-induced excitation from the ground state to an excited state, triggering rotation around the central C=C double bond and resulting in *E/Z* isomerization. In the subsequent THI step, the molecule undergoes a thermally activated helical flip, driven by steric strain and thermal energy.

We have added the data and the detail of DFT calculations in the Supplementary Information (Page 19, Supplementary Fig. 26) and these descriptions in revised manuscript (Page 3, line 74-78 and Page 5, line 164-166).

Fig. R8 (Supplementary Fig. 26) Optimized structure and dipole moments of (a) *Z_s*-2MOEG4 and (b) *E_s*-2MOEG4.

3. Why the authors chose second-generation molecular motor amphiphiles?

The second-generation motor was chosen because its rotation speed (minutes for THI at room temperature) is compatible with dynamic LLPS regulation. We have added revised the related sentence to be more clear (Page 3, line 78-81).

4. The scheme representing supramolecular assemblies in Figure 1 is confusing and misleading. Are they twisted nanofibrils or homogeneous aqueous solutions? Please replace identical expressions and avoid misleading, or the authors may provide CD spectra or AFM as evidence of helical assembly.

We apologize for the confusion. No helical structure was observed. Cryo-TEM showed worm-like micelles, and CD spectra confirmed no supramolecular chirality.

The cartoon in Fig. 1 has been revised to reflect this more accurately.

Fig. R5 (Supplementary Fig. 27) CD and UV-vis absorption spectra of **Z₅-2MEG4** assembly (1mg/mL) in water in a cuvette with 1mm light pathlength.

5. According to monomer rotation studies, NMR and UV-Vis are carried out under distinct temperature treatment, this is confusing. Is the THI steps primary a result of temperature or time-dependent dynamic relaxation?

THI is a first-order time-dependent process, but its rate is strongly temperature-dependent. We used lower temperatures in NMR to stabilize metastable isomers during the measurements, and higher temperatures in UV-Vis to monitor kinetics in accessible time windows. This has been clarified in the revised Methods section (Page 15, line 475-476).

6. The Cryo-TEM image of figure 2f is poor in quality and request optimization with higher magnification. Besides, please supplement experimental details in developing supramolecular assemblies.

We have optimized Fig. 2f with higher magnification. In addition, we have also replaced Fig. 2g with higher magnification version to have a better comparison with revised Fig. 2f. Detailed sample preparation protocols have been added to the Methods section (Page15, line 459-461).

7. According to thermal-responsive LLPS, how does stable E isomers be obtained? Why only stable E/Z isomers are discussed instead of metastable states? Additionally, please compare fluidity of ES-2MOEG4, ZS-2MOEG4, ES-2MOEG6 and ZS-2MOEG6.

We thank the reviewer for these insightful questions and suggestion.

Stable E isomers are obtained through a photoisomerization followed by THI. The thermal-responsive LLPS studies focused on the stable E/Z isomers (ES and ZS) because the metastable isomers (EM and ZM) are thermally unstable. They spontaneously undergo THI with half-lives ranging from minutes to hours, depending on temperature. making it experimentally difficult to isolate metastable isomers and study their thermal-responsive LLPS.

To assess droplet fluidity, we attempted FRAP measurements on all four assemblies. However, only **Z₅-2MOEG4** formed large and stable enough droplets at moderate temperatures for reliable photobleaching and recovery analysis (Fig. 3h). The other isomers underwent LLPS at higher temperatures, where the resulting droplets were small (300–400 nm) and highly mobile. These conditions made it technically difficult to accurate bleach the dye in droplets and perform FRAP due to rapid droplet movement and limited spatial resolution.

To indirectly compare the fluidity of the four types of droplets, we investigated the rotational speed of the molecular motors within them. Since the rotation speed of metastable isomer to stable isomer is influenced by the surrounding environment, lower fluidity may result in slower motor rotation. We performed Eyring analysis of THI process of **2MOEG6** (Fig. R9 and R10) and compared with that of **2MOEG4** within the droplets (Supplementary Figs. 22 and 23). The critical phase separation temperatures of **2MOEG6** and **2MOEG4** differ; therefore, measurements were conducted at temperatures that allow LLPS for each molecule. For a fair comparison, all half-lifetimes were calculated at 20 °C; however, these values still reflect the motor rotation speed in their measurement environments. The half-lifetimes of E_M -**2MOEG6** and Z_M -**2MOEG6** at 20 °C are 1.1 minutes and 1.3 minutes, respectively, which are significantly shorter than those of E_M -**2MOEG4** and Z_M -**2MOEG4** (7.4 minutes and 9.3 minutes, respectively, Table R1). These results suggest **2MOEG6** droplets exhibit higher fluidity than those of **2MOEG4**, possibly due to weaker hydrogen bonding and looser molecular packing at the higher temperatures required for LLPS.

These data and the related discussion have been included in the revised Supplementary Information (Supplementary Figs. 24 and 25; Supplementary Table 1; Page 20, Discussion 2).

Fig. R9 (Supplementary Fig. 24) (a) Absorption at 440 nm subtracting the absorption at 500 nm as a function of time during the THI of E_M -**2MOEG6** to form E_S -**2MOEG6** in water after phase separation (in the droplets) at different temperatures. (b) Eyring analysis on the THI after phase separation in water.

Fig. R10 (Supplementary Fig. 25) (a) Absorption at 440 nm subtracting the absorption at 500 nm as a function of time during the THI of Z_M -2MOEG6 to form Z_S -2MOEG6 in water after phase separation (in the droplets) at different temperatures. (b) Eyring analysis on the THI after phase separation in water.

Table R1. Half-life time of metastable motors in droplets.

System	$t_{1/2}$ (min) ^a $E_M \rightarrow E_S$	$t_{1/2}$ (min) $Z_M \rightarrow Z_S$
2MUOEG4 in droplets	7.4	9.3
2MUOEG6 in droplets	1.1	1.3

^aHalf-life ($t_{1/2}$) defined as $\ln(2)/k$ at 20 °C.

8. The discussion of LLPS during rotary motion is controversial. For example, Figure 4a makes no significant trend of full-spectrum absorption and more representative tests are request. Figure 4i reflected negligible LLPS while the author claimed “the solution of ZS-2MOEG4 supramolecular assemblies underwent LLPS to form droplets (Fig. 4i and k)”. Additionally, the conclusion at the end of first paragraph stated that reversible LLPS can be induced under $T_c(Z) < T < T_c(E)$, which is contradictory to general trend $T_c(E) < T_c(Z)$. Please clarify the above discrepancy and revise your article carefully.

In Figure 4a, the modest increase in absorption at 500 nm reflects LLPS-induced scattering. The signal is weak due to small droplet size (~490 nm), confirmed by DLS (Supplementary Fig. 9).

Fig. 4i shows the solution before LLPS, and Fig. 4i shows the droplets after LLPS. We appologies for the less clear description. We have corrected the text to reflect this (Page 8, line 233-234).

The T_c trend was misstated in this sentence. The correct relation is $T_c(E) < T_c(Z)$, and this has been corrected in the revised manuscript (Page 8, line 228-229).

9. Why both energy barrier and half-lifetimes of assemblies slightly decreased after phase separation.

We thank the reviewer for this insightful question. The observed slight decrease in the energy barrier and half-lifetime of motor rotation after phase separation is likely due to changes in the internal packing environment within the LLPS droplets.

Prior to LLPS, the supramolecular assemblies are stabilized by strong hydrogen bonding and π - π stacking interactions between aromatic motor cores. These interactions lead to relatively tight molecular packing, which restricts the rotational freedom of the motors and contributes to higher activation barriers (*Angew. Chem. Int. Ed.* **2024**, 63, e202319387).

However, LLPS typically occurs at elevated temperatures, which can weaken hydrogen bonding and loosen supramolecular packing. In the condensed phase (droplets), the internal environment is less tightly packed and more dynamic. This provides greater free volume and lower interaction constraints, allowing the molecular motors to rotate more freely. Therefore, the slight reduction in both energy barrier and half-lifetime observed in droplets is consistent with a more flexible, less densely packed local environment at elevated temperatures.

We have included this mechanistic explanation in the revised Supplementary Information (Page 20, Discussion 1).

10. Whether two-modal control can be orthogonal coupled for LLPS regulation?

We appreciate the reviewer for raising this point. Orthogonal two-modal control refers to a strategy in which two distinct external stimuli independently regulate different molecular or material properties within a single system. In alignment with this definition, our study has demonstrated orthogonally coupled regulation of LLPS: photoisomerization is driven by light, while thermal helix inversion (THI) is governed by temperature, and these two processes occur independently. Furthermore, the LLPS behavior of the stable isomers is also regulated by temperature, allowing for additional tunability after isomerization.

To better illustrate this orthogonal control mechanism, we have reorganized Fig. 1 (Fig. R11) in the revised manuscript to explicitly highlight the orthogonal coupled roles of light and temperature in controlling LLPS behavior.

Fig. R11 (Fig. 1) Molecular motor-driven multi-state liquid-liquid phase separation of supramolecular assemblies. (a) Four-state rotary of second-generation molecular motor amphiphiles and (b) the corresponding light and temperature control of liquid-liquid phase separation based on an aqueous solution of the molecular motor assemblies.